# Juniper Berry Oil as a Functional Additive in Chitosan–Water Kefiran–Paramylon Porous Sponges: Structural, Physicochemical, and Protein Interaction Insights

**DOI:** 10.3390/ijms26115314

**Published:** 2025-05-31

**Authors:** Dorota Chelminiak-Dudkiewicz

**Affiliations:** Department of Biomedical Chemistry and Polymer Science, Faculty of Chemistry, Nicolaus Copernicus University in Torun, Gagarina 7, 87-100 Torun, Poland; dorotachd@umk.pl

**Keywords:** juniper oil, water kefiran, paramylon, biomaterials, biomedical application

## Abstract

This study reports on the design and development of novel porous biomaterials based on chitosan, water kefiran, and paramylon, enriched with various concentrations of juniper berry oil (JBO). The materials were obtained by freeze-drying and comprehensively characterized. The analyses included morphological evaluation (SEM and porosity), physicochemical tests (swelling rate, water vapor transmission rate, and roughness), mechanical tests (tensile strength, Young’s modulus, and elongation at break), and biodegradability under physiological conditions. Moreover, the functional behavior of the materials was evaluated by assessing their antioxidant and anti-inflammatory activity, as well as interactions with selected proteins (human serum albumin and fibrinogen) relevant to biological responses. It was found that the presence of JBO affects the internal structure and improves selected properties in a concentration-dependent manner. This study is the first to investigate the combined use of chitosan, water kefiran, and paramylon in a single porous system enriched with JBO. The results confirm the importance of such biopolymer sponges as promising platforms for applications where appropriate physicochemical and bioactive properties are desired.

## 1. Introduction

Biomaterials are materials produced to work in contact with biological systems and are used in medical, industrial, and environmental sciences [1,2,3]. These properties make them necessary for producing regenerative therapies, precise drug delivery systems, environmentally friendly material development, and water filtration devices [4,5,6,7,8]. Due to the variety of applications, biomaterials can be formed into films, foams, nanofibers, hydrogels, and porous sponges [9,10,11,12,13]. Among these, sponges based on biopolymers have recently attracted much attention. They are also well known for their use in wound healing, drug delivery, and tissue engineering due to their high porosity [14,15,16]. The complex porous structure improves and thus regulates the transport of fluids and cellular migration and diffusion of bioactive molecules, making them a vital material in developing advanced wound care and tissue engineering devices [17,18].

Over the past few years, many studies have been conducted to develop new biopolymer sponges with better performance. Using natural polysaccharides results in the formation of highly biocompatible and biodegradable materials. Chitosan (CS), among the various polysaccharides, is widespread. CS is a cationic polymer obtained from chitin, the most common component of the exoskeleton of arthropods [19,20]. Because of its specific features, such as biocompatibility, biodegradability, eco-friendliness, hydrophilicity, high adsorption capacity, and low cost, it has been used in biomedicine [21]. However, pure chitosan has some limitations, including a low adsorption rate and poor mechanical strength [21]. To overcome these disadvantages, CS is usually combined with other polysaccharides.

Of all the combinations, kefiran is quite remarkable. It is an exopolysaccharide produced by lactic acid bacteria in kefir grains in lactic fermentation. Structurally, kefiran consists of D-glucose and D-galactose in a chain sequence [22]. It has antioxidant and antibacterial activities and prebiotic and immunomodulatory effects [23,24]. Previous studies have been carried out mainly on kefiran isolated from milk kefir grains, and there are promising opportunities for its application in different fields [25,26]. However, water kefir (WK) may have the same physicochemical and bioactive properties and can be used for the same goals. Thus, it was used in this study.

In addition to kefiran, paramylon (P) was also used. It is a linear β-1,3-D-glucan that serves as a storage polysaccharide in *Euglena gracilis* cells [27]. This unique structure makes it chemically stable and resistant to enzymatic degradation, as opposed to the normal β-glucans obtained from yeast, oats, barley, and wheat [28]. Due to these characteristics, paramylon is currently being considered for use in the food industry, for environmental protection, and as a filler in biodegradable polymer composites [29,30]. However, there is a lack of information on using paramylon in porous structures such as biopolymer sponges, although research on this polysaccharide is increasing. Previous work has been mainly directed towards the biological functions of the polymer, while its capability to generate porous materials has not been investigated. The situation is similar for water kefiran, a promising source of bioactive compounds which has not been investigated as a component of biopolymer sponges.

Essential oils have antimicrobial, antioxidant, and anti-inflammatory activities, and many works have found that incorporating oils into biopolymers enhanced bioactivity and improved the physicochemical properties of the materials. Despite the growing interest in using essential oils in biomaterials, not all essential oils have been investigated for incorporating into biopolymer sponges in detail. One is juniper berry oil (JBO), which has not yet been used in porous structures. JBO is obtained from the fruits and needles of *Juniperus communis* and contains monoterpenes (α-pinene, β-pinene, limonene, and sabinene), sesquiterpenes (caryophyllene and germacrene D), flavonoids, and caffeic acid. It exhibits antimicrobial, antioxidant, and anti-inflammatory properties [31,32].

This work aimed to develop and investigate sponges based on a three-component system of polysaccharides (chitosan, water kefiran, and paramylon) added with juniper oil. The influence of different oil concentrations on the structure and physicochemical properties of the obtained material was also investigated. The structure and morphological characteristics (ATR-FTIR, SEM, and porosity), absorption properties (swelling rate and WVTR), biodegradation, mechanical properties, and antioxidant and anti-inflammatory activity were characterized. In addition, the interaction of the sponges with human serum albumin (HSA) and fibrinogen was investigated to determine their potential biomaterial functions. Sponges based on this combination of polysaccharides with the addition of juniper oil have not been described to date. The combination of these components can influence bioactivity, mechanical properties, and interactions with proteins, which offers new possibilities in biomaterial engineering.

## 2. Results and Discussion

### 2.1. Preparation and Characterization of the Chitosan–Water Kefiran–Paramylon Sponges Enriched with Juniper Berry Oil (CS/WK/P%JBO)

Polysaccharide-based materials are a good alternative to synthetic materials, especially in biomedical applications. They are non-toxic and compatible with the body [33,34,35,36,37,38,39,40,41,42,43,44,45]. Therefore, it was decided to use them to design new materials. However, using single polysaccharides has some disadvantages; thus, it is better to design compound systems. Consequently, it was decided to use a mixture of chitosan, water kefiran, and paramylon for this work. Additionally, juniper berry oil was used to improve the properties of such a material, which is known for its anti-inflammatory and antioxidant properties [36,37,38,39]. The obtained materials are in the form of a sponge. This structure is ideal for biomedical applications, especially wound dressing materials, due to its high porosity and ability to absorb liquids, promoting gas exchange, bioactive substances’ transport, and tissue integration. The scheme for obtaining the porous material is shown in Figure 1.

The morphology of the obtained sponges was observed using scanning electron microscopy (SEM) and an optical microscope. The SEM images (Figure 2A) show the oil’s apparent influence on the samples’ morphology. The CS/WK/P sponge is more compact and smoother than the other samples. Furthermore, the porosity analysis (Figure 2B), examined with an optical microscope, showed that this sample has a porosity of 41%. This is consistent with the test carried out using absolute ethanol (41 *±* 1.2%). Adding JBO oil increased the porosity, with the CS/WK/P5% sponge showing the smallest increase. The highest porosity was demonstrated by the CS/WK/P20% sample (73.5%). This value is close to the result obtained in the study using absolute ethanol (74.12 *±* 1.09%). The increase in porosity with increasing oil concentration may be due to the disruption of the biopolymer structure during sponge formation. The presence of oil promoted the formation of larger and more numerous pores, acting as a pore-forming agent. This phenomenon is consistent with the literature. In their study, Saranti et al. [40] stated that adding a black pepper essential oil nanoemulsion increased the gelatin matrix’s porosity. The surface roughness of biomaterials is crucial for their functionality, especially regarding adhesion, fluid absorption, and interactions with cells and proteins [41,42]. This study showed that adding juniper oil significantly affected the surface topography of biopolymer sponges. The analysis of roughness parameters showed that R_max_ increased with the concentration of the oil—from 72.14 µm for the CS/WK/P to 131.39 µm and 144.27 µm for CS/WK/P5%JBO and CS/WK/P5%JBO, respectively, up to 177.81 µm at 20% oil content. This increase can be attributed to the presence of hydrophobic terpene compounds, which affect the drying process and the structuring of the surface during the formation of a porous matrix. The literature supports this [43,44]. Higher roughness can positively impact the material’s adhesive and absorption properties, as confirmed by the results of the adhesive force measurements. Changes in the microstructure of the samples were also confirmed by analyzing the three-dimensional surface profile. The 3D profile images clearly show an increase in topographical irregularity with increasing oil content, consistent with the results of the quantitative measurements. This structure could potentially promote increased contact with the substrate or wound surface, which is desirable for biomaterials with bioactive effects.

The presence of juniper berry oil (JBO) in the structure of the obtained materials was confirmed by ATR-FTIR spectroscopy (Figure 3A). The spectra of the polysaccharide mixture (CS/WK/P) and JBO are shown in the Appendix A (Appendix A). The CS/WK/P spectrum (Appendix A) shows characteristic bands for each of these polysaccharides, including O–H and N–H (3272 cm⁻^1^), C–H (2918 cm⁻^1^), and C=O/N–H (1636–1407 cm⁻^1^) bands and intense C–O–C and C–OH vibrations in the 1200–800 cm⁻^1^ range [45,46]. Moreover, a band at 883 cm⁻^1^ confirms the presence of paramylon (β-glucan) [47], and the broad and intense bands indicate interactions between the mixture’s components. In the spectrum of juniper oil (Appendix A), the broadband at around 3430 cm⁻^1^ is attributed to the stretching vibrations of O–H groups originating from phenolic components [48]. The intense bands at 2923 cm⁻^1^ and 2856 cm⁻^1^ correspond to the stretching vibrations of C–H in methyl and methylene groups, characteristic of aliphatic chains of monoterpenes and sesquiterpenes [49,50]. The band at 1741 cm⁻^1^ is related to the stretching vibrations of carbonyl groups (C=O) of esters [51,52], while the band at around 1370 cm⁻^1^ is associated with the bending vibrations of C–H in aliphatic structures [53]. The spectrum of the obtained sponges shows bands of 2800–2930 cm^−1^ and a band of 1742 cm^−1^ [49,50], confirming the successful introduction of the oil into the polysaccharide matrix. However, these bands differ in intensity, which indicates an increasing proportion of oil components in the sponges.

Mechanical properties are crucial for the evaluation of the suitability of new biomaterials. Therefore, the tensile strength, elongation, and Young’s modulus of the obtained sponges were examined, and the results were compared with materials without oil (Table 1). The CS/WK/P sponge without oil had the lowest tensile strength (TS = 2.25 ± 0.02 MPa), elongation (5.0 ± 0.3%), and the highest stiffness (Young’s modulus: 57.2 ± 2.4 MPa). With increasing JBO oil content, a clear improvement in mechanical properties was observed: for the CS/WK/P5% sample, the tensile strength increased to 2.39 ± 0.04 MPa, elongation to 9.2 ± 0.2%, and the modulus decreased to 48.1 ± 1.9 MPa. The best combination of strength and elasticity was obtained for the CS/WK/P10%JBO sample (TS = 2.88 ± 0.03 MPa, strain = 13.5 ± 0.4%, Young = 39.7 ± 2.1 MPa). In the CS/WK/P20% sponge, the parameters remained promising (TS = 2.58 ± 0.05 MPa, strain = 18.0 ± 0.6%, Young = 28.1 ± 1.7 MPa), indicating a further increase in elasticity at the expense of a slight decrease in strength. The increase in elongation with a simultaneous decrease in Young’s modulus and maintenance of high strength indicates an improvement in the plasticity of the structure. The observed changes are also confirmed at the microstructural level. Surface roughness analysis showed a significant increase in the R_max_ parameter in samples containing JBO (from 72.14 µm for CS/WK/P to 177.81 µm for CS/WK/P20%). The varied topography promotes stress dispersion and reduces the risk of crack initiation, which supports the improved resistance to brittle damage and enables a more even transfer of mechanical loads within the structure. Furthermore, the presence of hydrophobic JBO components may limit excessive hydration, and its known plasticizing effect increases the mobility of polymer chains, which also contributes to reducing brittleness and increasing the deformability of the material. These studies are consistent with the literature reports. Biswas et al. [54] investigated the impact of essential oils on the properties of cellulose film. They showed that the elongation at break increased significantly for films containing five essential oils (lime, nutmeg, eugenol, aniseed, and trans-cinnamaldehyde). Another study by Boro et al. [55] found that adding clove oil to PLA nanocomposites reduced the Young’s modulus and tensile strength while increasing the material’s elasticity. Based on the results of the mechanical properties, it can be concluded that the obtained CS/WK/P%JBO sponges show promising potential for applications in biomedical engineering, where adequate durability and soft tissue adaptation are required.

Similar findings have been reported in the literature. Yeşilyurt et al. [56] demonstrated that PVA/κ-carrageenan films containing juniper oil exhibit a significant increase in elongation at break with moderate tensile strength (TS = 5.65 MPa), which the authors attributed to the action of JBO as a natural plasticizer. In another study, the authors used *J. communis* extract in a Poly(Butylene Adipate-Co-Terephthalate) matrix and found improved mechanical properties [57]. The authors demonstrated that the film’s stiffness was reduced, and its elongation improved. Bhatia et al. [58] also reported a similar effect in a study of starch–pectin films. The addition of JBO resulted in a systematic increase in elongation at the break—from 13.51% (control film) to 18.53% in the sample containing the highest oil concentration. The increased elasticity was attributed to the presence of the hydrophobic components of the oil, which reduced brittleness and increased the freedom of movement of polymer chains.

Cyclic tensile tests were done to assess the materials’ usefulness for potential biomedical applications, recording the maximum stress and hysteresis energy as mechanical resistance and durability indicators under dynamic loading. These tests were performed in a system comprising 10 consecutive tensile–tempering cycles, with no time interval between cycles. This number of cycles is consistent with the literature evaluating polymer and composite materials used in biomedical engineering [59,60]. The results are summarized in Figure 4 and Appendix A. All obtained samples showed characteristic hysteresis loops, confirming the involvement of energy dissipation during deformation. The CS/WK/P sponge significantly reduced maximum stress (−18%) and hysteresis energy (−18%) between cycles 1 and 10, revealing its poor fatigue resistance.

Adding juniper berry oil enhanced the material’s mechanical properties at all concentration levels. At a JBO content of 5%, an increase in initial stress and energy was observed compared to the control sample. However, the decreases after 10 cycles remained comparable (−19%), suggesting a partial improvement in mechanical properties but without complete protection against fatigue. The best properties were exhibited by the material containing 10% JBO (CS/WK/P10%JBO), which was characterized by the highest initial strength (2.88 MPa) and dissipated energy (86.4 kJ/m^3^), and after 10 cycles, these values decreased only slightly (−6% stress, −5.7% energy). Such a low level of degradation confirms the high resistance of the material structure to repeated mechanical loads. In the case of the CS/WK/P20%JBO sponge, favorable initial values were observed. However, the parameter decrease after 10 cycles was more pronounced (−16% stress, −17% energy), which may suggest excessive plasticization of the polymer network or a supersaturation effect.

The results of the adhesion force test (Figure 3B) showed a statistically significant increase in the adhesion of the obtained sponges with an increase in JBO concentration, regardless of the type of substrate. On the surface of the glass, the adhesion strength increased from about 1.0 N for CS/NK/P to about 2.7 N for the CS/NK/P20% sponge. In the case of metal, an increase from about 1.5 N to over 3.5 N was observed, and for rubber, from 2.0 N to nearly 4.5 N. For plastic surfaces, the adhesion strength increased from approximately 1.4 N for the CS/WK/P sample to approximately 3.6 N for CS/WK/P/20%JBO. However, the highest adhesion values were observed for pig skin. The force increased from approximately 2.1 N to over 5 N in this case. Such a marked improvement may result from the greater susceptibility of the soft, moist skin surface to contact with the elastic, hydrophobic material and from the contribution of the oil components to the formation of better surface interactions. The results indicate that adding JBO improves the material’s adhesive properties, which can be important for applications requiring strong, selective adhesion to different types of surfaces (such as wound dressing materials).

### 2.2. Water Vapor Transmission Rate (WVTR)

The water vapor transmission rate (WVTR) is essential for assessing biopolymer materials’ functionality. A suitable WVTR level controls moisture exchange between the material and the environment. A systematic increase in WVTR values was observed in the analyzed samples as the concentration of juniper oil increased (Table 2). The CS/WK/P sample reached a WVTR of 1950 ± 60 g·m^−2^·day^−1^, while adding the oil increased these values. The highest WVTR value was observed for the CS/WK/P20% sponge. This trend may be directly related to the increase in porosity and roughness of the structure, which promotes more effective diffusion of water vapor through the material. Increased WVTR indicates an improvement in the transport properties of materials, which may benefit their further applications, such as wound dressing materials. Many reports in the literature confirm the positive effect of adding essential oils. In a study based on a mixture of chitosan, gelatin, and pectin with rosemary oil added, an improved WVTR was observed [61]. The same was true in the case of a study involving the addition of orange oil [62]. A broader comparison of WVTR values from the literature and those obtained in this study is presented in Table 2. It is worth noting that the CS/WK/P20%JBO sponge showed a WVTR of 2250 ± 70 g·m^−2^·day^−1^, which exceeds many values reported for polysaccharide-based biomaterials, reflecting its increased ability to transport water vapor through the material.

### 2.3. Swelling Properties

Biomaterials swell because of their internal structure, porosity, and ability to form water bonds. This property is essential for materials in contact with moist biological environments, such as wound dressing materials [68]. The results of the swelling test were carried out on a sponge without added juniper berry oil and with different concentrations of juniper berry oil in the sample (Figure 5A). An apparent correlation between swelling capacity and JBO concentration was observed. Of all the samples, the CS/WK/P sponge showed the lowest swelling (496 ± 6.08% after 24 h), while the CS/WK/P20% sponge had the highest value (995 ± 5.03% after 24 h). These results are directly related to the porosity and roughness of the samples. The sponges’ porosity and roughness obtained rise with the JBO concentration. This might lead to better penetration and retention of water in the material structure. In addition, according to the Young’s modulus measurements, the material becomes more elastic and less rigid as the oil concentration increases, which promotes fluid absorption. Similar effects were observed in other studies. In research on alginate hydrogels containing Mentha (*Mentha arvensis*) essential oil loaded in chitosan nanoparticles, it was observed that the oil enhanced the hydrogels’ porosity, resulting in an increased swelling capacity [69]. Another study analyzed the effect of adding *Citrus limonia* essential oil to chitosan films. It was found that the presence of the oil increased the material’s swelling capacity [70]. The improved swelling capacity with the increasing concentration of juniper oil indicates beneficial modifications to the biomaterial’s structure, which may promote its more effective functioning in a moist environment.

### 2.4. Biodegradation Ratio

The decomposition rate of materials in biological environments establishes their practical use and safety for biomedical applications [71]. The obtained materials underwent a biodegradation test for 14 days (Figure 5B). The research results demonstrated that increasing juniper berry oil content enhances the biodegradability of sponge structures. The CS/WK/P20%JBO sample reached the highest biodecomposition ratio—nearly 80% after 14 days, while the CS/WK/P sponge decomposed much more slowly (approx. 63%). This effect may be related to the increase in porosity and roughness of the material, which facilitates the access of enzymes and water to its interior, accelerating the decomposition of the polymer structure. In the literature are reports on the impact of adding essential oils on the biodegradation of biomaterials used in medicine. An example is a study conducted by Hussein et al. [72], which analyzed nanofibrous polyurethane (PU) and PVA–gelatin scaffolds containing cinnamon oil (CEO) and cerium nanoparticles (nCeO_2_). The results showed that the presence of the CEO initially slowed down the degradation of the materials. However, the materials’ structures destabilized as the oil was released, increasing the degradation rate. After 14 days, the samples with CEO showed higher biodegradation than the control samples.

### 2.5. Anti-Inflammatory Properties

The evaluation of the anti-inflammatory properties of biomaterials is essential because chronic inflammation may delay wound healing and the function of implants [73]. Thus, natural substances with therapeutic properties are being researched. It is worth mentioning Juniper berry oil because it contains, among other things, the monoterpenes α-pinene and β-pinene, which have anti-inflammatory effects [74].

This study revealed that increasing the JBO concentration in the biomaterial matrix led to substantial growth in anti-inflammatory properties (Figure 5C). The CS/WK/P sponge exhibited the minimum inhibition of BSA denaturation, but the CS/WK/P/20% sponge reached more than 60% inhibition when tested at 100 µg/mL. The 5% addition of essential oil caused a slight increase in protein denaturation inhibition. However, a further rise in JBO concentration (10 and 20% by weight relative to polysaccharides) significantly improved the anti-inflammatory properties. At the highest concentration (500 µg/mL), CS/WK/P10% achieved 68.83 *±* 0.76%, and CS/WK/P20% 81.7 *±* 2.51%, close to the model compound diclofenac sodium. According to these findings, adding JBO to biomaterials produces substantial anti-inflammatory effects, making the material suitable for medical applications requiring inflammation control during healing. Significantly, JBO inhibited protein denaturation in a concentration-dependent manner, achieving over 50% inhibition at 500 µg/mL. While diclofenac sodium, a synthetic anti-inflammatory agent, consistently exhibited higher inhibition values across all concentrations, the biological effect of JBO was clearly evident and statistically significant (****, *p* < 0.0001) (Appendix A).

### 2.6. Antioxidant Properties

Oxidative stress may significantly affect the functionality of biomaterials used in regenerative medicine. Therefore, their ability to neutralize free radicals is a highly biologically and clinically relevant parameter [75]. This study observed an apparent increase in the DPPH free radical scavenging capacity with increasing juniper oil concentration in the CS/WK/P matrix (Figure 5D). Even at the lowest concentration tested (10 µg/mL), JBO-containing sponges showed a significantly higher level of free radical scavenging than CS/WK/P. The %DPPH values increased gradually from CS/WK/P5%JBO through CS/WK/P10%JBO to CS/WK/P20%JBO, reaching 16 *±* 0.90%, 19 *±* 0.96%, and 23 *±* 0.99%, respectively, compared to 9 *±* 1.15% for the CS/WK/P sample. This trend persisted in all analyzed concentrations up to 500 µg/mL, where the CS/WK/P20% sample showed DPPH free radical scavenging at 85 *±* 1.28%, approaching the values achieved by ascorbic acid. The synergistic combination of the polymer matrix and active components of the essential oil leads to this observed effect. The oil contains α-pinene, limonene, and sabinene, which serve as documented antioxidants according to the scientific literature. Moreover, JBO showed a progressive increase in radical scavenging ability, reaching approximately 45% at 400 µg/mL and 52% at 500 µg/mL, which means the antioxidant potential of the oil remains notable and confirms its role as a natural radical scavenger. The morphological analysis results show that the structure’s biomaterial porosity and roughness are crucial in enhancing this effect. The material’s contact surface area increases with its active ingredient diffusion, leading to better DPPH radical interactions. The swelling capacity positively correlates with the migration of soluble substances like terpenes from the structure’s interior to the environment.

Similar findings have been observed in other studies. Hu et al. [76] analyzed essential oils from different juniper species, and they found Juniperus sabina oil has a strong antioxidant effect due to its high sabinene content.

### 2.7. Protein Adsorption

When biomaterials come into contact with biological fluids, proteins begin to adhere almost instantly to their surface. This initial interaction can substantially impact various cellular processes, such as blood clotting, immune response, and tissue healing [77,78]. The albumin human serum (HSA) and fibrinogen (Fib) protein adsorption efficiencies of the obtained sponges are shown in Figure 6A. The results clearly show that the amount of adsorbed HSA increased proportionally to the concentration of JBO oil. After 24 h, the CS/WK/P20%JBO sponge showed the highest adsorption (2.30 *±* 0.02 mg/cm^3^), significantly exceeding CS/WK/P, which exhibited the lowest values (0.57 *±* 0.01 mg/cm^3^). Noteworthy is the rapid increase in HSA adsorption already after 60–120 min for samples containing JBO, suggesting that the presence of oil increases the affinity of the protein to the surface of the material. This effect can be attributed to the higher porosity and roughness of the surface of the modified materials, which provide an expanded area for protein anchoring. Additionally, the increased swelling capacity observed for JBO-enriched sponges may facilitate protein penetration into the matrix. At the same time, the chemical components of the oil, rich in polar and aromatic compounds, may further contribute through hydrogen bonding and electrostatic interactions with serum proteins.

A similar trend was observed for fibrinogen adsorption, where the CS/WK/P20%JBO sponge bound the most protein (2.14 *±* 0.09 mg/cm^3^ after 24 h) on its surface, in contrast to the CS/WK/P sponge (0.90 *±* 0.04% mg/cm^3^). Since Fib is a key protein in the coagulation cascade and modulates early inflammatory responses [79], its increased adsorption indicates the potential of JBO-modified materials to enhance initial hemostatic activity.

## 3. Materials and Methods

### 3.1. Materials

Chitosan (low molecular weight, deacetylation degree of 79%), paramylon (β-1,3-glucan from *Euglena gracilis*), phosphate buffer saline (PBS) (pH 7.4), diclofenac sodium, human serum albumin (HSA), bovine serum albumin (BSA), lysozyme, and 2,2-diphenyl-1-picrylhydrazyl radical (DPPH) were purchased from Sigma-Aldrich (Munich, Germany). Japanese algae to produce water kefiran were purchased from the local culinary store. Acetic acid (99.8%), hydrochloric acid, and phenolphthalein were purchased from Avantor Performance Materials Poland (Gliwice, Poland). Juniper berry oil (Etja) was purchased from a local pharmacy. Water was purified with a Milli-Q Water Purification System (Millipore Corp., Bedford, MA, USA).

### 3.2. Kefir Grain Growth

Water kefir grains (WK) were grown according to the method used by Coma et al. [80], with some modifications. The Japanese algae (20 g) were washed in a plastic sieve under tap water. Then, they were put into a glass jar (1 L) with 20 g of sugar, 15 g of dried figs, and two slices of lemon. A total of 1 L of freshly filtered water was poured into the jar, which was then covered with a cloth and left for 72 h. After this time, the algae were separated from the fermented product by filtering through a plastic sieve, washed two times with distilled water, and squeezed to remove excess water. The grains were dried in an oven at 90 °C for 12 h and then ground into powder in a blender.

### 3.3. Preparation of the Chitosan/Water Kefiran/Paramylon Sponges Enriched with the Juniper Berry Oil (CS/WK/P%JBO)

First, the polysaccharide solutions were prepared. For this purpose, 0.1 g of the respective polysaccharides (chitosan, water kefiran, and paramylon) were weighed into separate flasks. A total of 10 mL of acetic acid (C = 1%) were added to each flask. Each solution was mechanically stirred at room temperature for 1 h. Then, the prepared solutions were mixed and stirred for 1 h at room temperature. Next, juniper oil (1.0% by weight to the polysaccharide mixture) was added and stirred for 30 min. The mixture was poured into a Petri dish (90 mm in diameter), frozen (−20 °C, 8 h), and freeze-dried for 24 h in a freeze-dryer (−53 °C, 0.05 mBar).

Sponges with juniper berry oil in higher concentrations (10.0% and 20.0% by weight to the polysaccharide mixture) were obtained using the same method. The groups of sponges obtained were designated as CS/WK/P%1JBO, CS/WK/P%10JBO, and CS/WK/P%20JBO, respectively, for 1%, 10%, and 20% juniper berry oil content.

### 3.4. Characterization of Chitosan/Water Kefiran/Paramylon Sponges Enriched with Juniper Berry Oil (CS/WK/P%JBO)

#### 3.4.1. Attenuated Total Reflectance Spectroscopy (ATR-FTIR)

Structures of the mixture of chitosan, water kefiran, and paramylon, juniper berry oil, and CS/WK/P%JBO sponges were characterized by the Attenuated Total Reflectance Fourier Transform Infrared (ATR-FTIR) spectroscopy with the Spectrum Two™ (Perkin Elmer, Waltham, MA, USA) apparatus with a diamond crystal. Spectra were recorded in the range of 4000 to 450 cm^−1^ at a resolution of 16 cm^−1^ (32 scans). After recording the spectra, the baseline and ATR corrections were made.

#### 3.4.2. Scanning Electron Microscopy (SEM)

The morphology of the obtained sponges was studied with a Scanning Electron Microscope (1430 VP LEO Electron Microscopy Ltd, Cambridge, UK). For best viewing under a microscope, the samples were coated with a gold layer.

#### 3.4.3. 3D Profile and Surface Roughness

Surface roughness (3D) analysis was performed using a digital microscope (Keyence VHX-X1, Keyence Corp., Osaka, Japan). Three-dimensional images of the surface were acquired at 100× magnification. Roughness parameters, including arithmetic average roughness (R_a_) and mean peak-to-valley height (R_z_), were calculated using the integrated analysis software. For each sample, measurements were taken from three different randomly selected areas.

#### 3.4.4. Porosity

The CS/WK/P%JBO sponges were dried at 50 °C for 2 h in a vacuum oven. Then, the weighted sponges (0.2 g) were immersed for 4 h in absolute ethanol (4 mL). Next, the swollen sponges were then blotted to eliminate the extra ethanol using filter paper and weighed. The porosity of the sponges was calculated according to Formula (1):(1)$$Porosity\;(\%)=\frac{\left({W-W}_{0}\right)}{\rho *V}*100\%$$
where *W* is the weight of the sponge after immersing in ethanol, *W*_0_ is the weight of the sponge before immersing in ethanol, *ρ* is the density of absolute ethanol, and *V* denotes the volume of the sponge [81].

The porosity was also determined using a Keyence VHX-X1 optical microscope.

#### 3.4.5. Mechanical and Adhesive Properties

The mechanical properties of the mixture of chitosan, water kefiran, and paramylon, juniper berry oil, and CS/WK/P%JBO sponges were tested by the EZ-Test E2-LX Shimadzu texture analyzer (Shimadzu, Kioto, Japan) at room temperature. Five sample strips of the sponges were cut and clamped between pneumatic grips. The tests were performed at an extension rate of 20 mm/min, and each sample was repeated three times.

Hysteresis behavior was evaluated through tensile loading–unloading experiments performed on sponge-like samples using a universal testing machine. Each sample was stretched to a predefined maximum strain of 50%, followed immediately by unloading to 0% strain. This loading–unloading cycle was repeated 10 consecutive times without time intervals between cycles to assess the material’s fatigue resistance.

A universal testing machine was also used to measure the adhesive strength of the sponges. Glass, metal, rubber, plastic (polypropylene), and pork skin surfaces were used for the tests. The pork skin was purchased at a local butcher. Each of the material were bonded with the sponges (10 mm × 10 mm) and compressed for several minutes at room temperature. The test was performed at an extension rate of 20 mm/min, and each sample was repeated three times.

#### 3.4.6. Swelling Analysis

The swelling ratio of the prepared sponges was calculated by incubating material sponges (each sponge weighed about 0.2 g) in PBS solution (4 mL) at pH 7.4 at 37 °C. After wiping excess water, the sponges were weighed regularly. This experiment was performed in triplicate. The swelling rate was calculated according to Formula (2):(2)$$Swelling\;rate(\%)=\frac{\left(W_{sponge}-W_{Dry}\right)}{W_{Dry}}*100\%$$
where *W_Sponge_* is the weight of the swollen sample, and *W_Dry_* is the weight of the dried sample.

#### 3.4.7. Biodegradation Analysis

Biodegradation analysis of the CS/WK/P%JBO sponges was determined by monitoring their weight changes for 14 days, according to previous reports [82]. A known weight of the sponges was incubated in PBS (pH = 7.4, 37 °C) containing lysozyme (0.5 mg/mL). The enzyme-containing PBS solution was replaced daily to maintain enzymatic activity and the physiological relevance of the degradation conditions. Immersed sponges were taken out daily and washed with deionized water. The biodegradability of the sponges was calculated according to Formula (3):(3)$$Biodegradation\;(\%)=\frac{\left(W_{0}-W\right)}{W_{0}}*100\%$$
where *W*_0_ is the initial weight of the sponge, and *W* is the final weight of the sponge.

#### 3.4.8. The Water Vapor Transmission Rate (WVTR)

The water vapor transmission rate (WVTR) was determined by fixing the sponge (0.2 g) onto the opening of a round plastic box with a diameter of 40 mm, which contained 5 mL of distilled water. The box was placed in an oven at 37 °C for 24 h, and the WVTR was calculated according to Formula (4):(4)$$WVTR\;(\frac{\frac{g}{m^{2}}}{h})=\frac{\left(\frac{\Delta w}{\Delta t}\right)}{A}$$
where $\left(\frac{\Delta w}{\Delta t}\right)$ is the slope of the plot, and *A* denotes the effective transfer area [83].

#### 3.4.9. Protein Adsorption

The protein adsorption on the obtained CS/WK/P%JBO sponges was determined by a fluorescence method using two proteins: human serum albumin (HSA) and fibrinogen (Fib), according to the previous report [13]. First, the solution of HSA (6.24 μM) and Fib (3.78 μM) in PBS (pH = 7.4; 50 mM) was prepared. Then, 0.36 cm^3^ of each type of CS/WK/P%JBO sponge was immersed in four milliliters of freshly prepared HSA or Fib solution and incubated using a thermomixer (36 °C and 600 rpm). Fluorescence spectra were recorded at an excitation wavelength of 280 nm using a Jasco FP-8300 spectrofluorometer (Jasco, Tokyo, Japan) at various intervals, ranging 290–400 nm and 300–500 nm for HSA and Fib, respectively. The spectral recording range was 285–500 nm, the scanning speed was 100 nm/min, and the Ex/Em bandwidth was 2.5 nm/5 nm.

#### 3.4.10. Antioxidant Activity

The DPPH radical scavenging assay was applied to measure the radical scavenging activity of the CS/WK/P%JBO sponges according to the previously reported method [14]. Ascorbic acid was used as a main standard compound. First, a freshly prepared DPPH reagent (1.0 mM in ethanol) was added to the ascorbic acid and sponges (30 μL), and the reaction was incubated at room temperature for 30 min in the dark. Then, the absorbance was recorded against a blank at 517 nm using a UV-1800 spectrophotometer (Shimadzu, Japan). The experiment was performed three times. The DPPH radical scavenging was calculated according to Formula (5):(5)$$DPPH\;scavenging\;(\%)=\frac{\left(A_{DPPH}-A_{sponge}\right)}{A_{DPPH}}*100\%$$
where *A_DPPH_* is the absorbance of the DPPH solution, and *A_sponge_* is the absorbance of the sponge.

#### 3.4.11. Anti-Inflammatory Activity

The anti-inflammatory activity of the CS/WK/P%JBO sponges was determined by the inhibition of bovine serum albumin (BSA) denaturation, as described previously [84]. First, the BSA solution (5 mL, 5%) was incubated with the sponge (10–500 μg/mL) at 37 °C for 15 min in a thermomixer (200 rpm). The sponge was then heated in a water bath at 70 °C for 5 min. The absorbances of the obtained blends and the diclofenac sodium solution were measured spectrophotometrically at 278 nm. The percentage of inhibition of denaturation was calculated according to Formula (6):(6)$$Inhibition\;(\%)=\frac{\left(A_{sponge}-A_{control}\right)}{A_{sponge}}*100\%$$
where *A_control_* is the absorbance of the control, and *A_sponge_* is the absorbance of the sponge.

#### 3.4.12. Statistical Analysis

The obtained data were reported as mean ± standard error of the mean. The data has been analyzed using one-way analysis of variance (ANOVA) using GraphPad Prism 10 (GraphPad Software, San Diego, CA, USA), followed by the Dunnett test (to compare each mean with the control mean) or the post hoc Tukey test (to compare each mean with each other mean) for multiple comparisons. The statistical difference of all tests was significant, considering *p* < 0.05.

## 4. Conclusions

This study demonstrated that the preparation of sponges based on biopolymers containing increasing concentrations of JBO leads to several beneficial modifications in the material properties. A systematic analysis showed that higher essential oil content increased porosity, swelling capacity, surface roughness of the sponges, and improved water vapor transmission rate. Tensile strength and elongation at break also increased with oil concentration, indicating that the materials became stronger and more flexible. At the same time, Young’s modulus gradually decreased, confirming greater flexibility at elevated oil levels.

Notably, materials enriched with juniper oil exhibited increased anti-inflammatory and antioxidant activity, attributed to the essential oil’s inherent properties and the sponge matrix’s more open structure. It was also found that adding oil facilitates more significant protein adsorption, probably due to increased surface area and altered surface chemistry. These physicochemical changes and improved degradability suggest that the developed biopolymer sponges may be promising candidates for various biomedical applications, particularly where dynamic interactions with the biological environment are required. However, further studies, including cytotoxicity assessment, will be necessary to evaluate their suitability for specific applications.

## Figures and Tables

**Figure 1 ijms-26-05314-f001:**
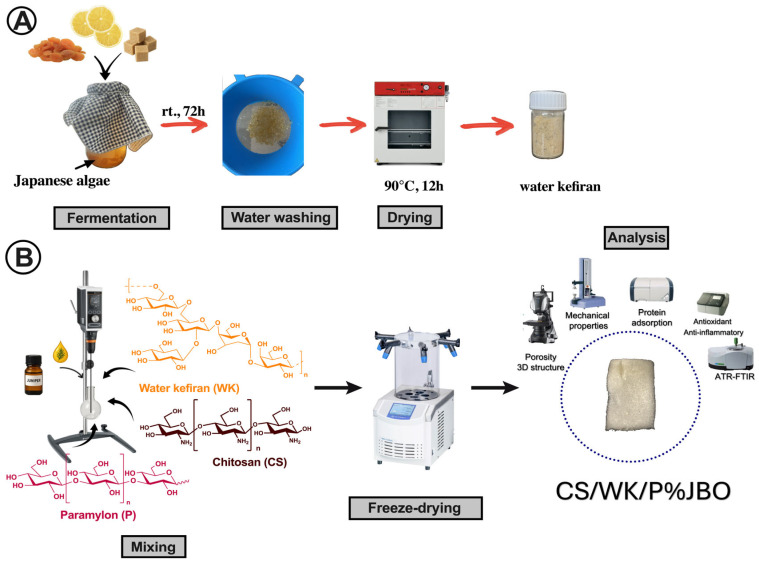
Graphic display of the main findings. Isolation of water kefiran (**A**) and scheme of obtaining CS/WK/P% sponges (**B**).

**Figure 2 ijms-26-05314-f002:**
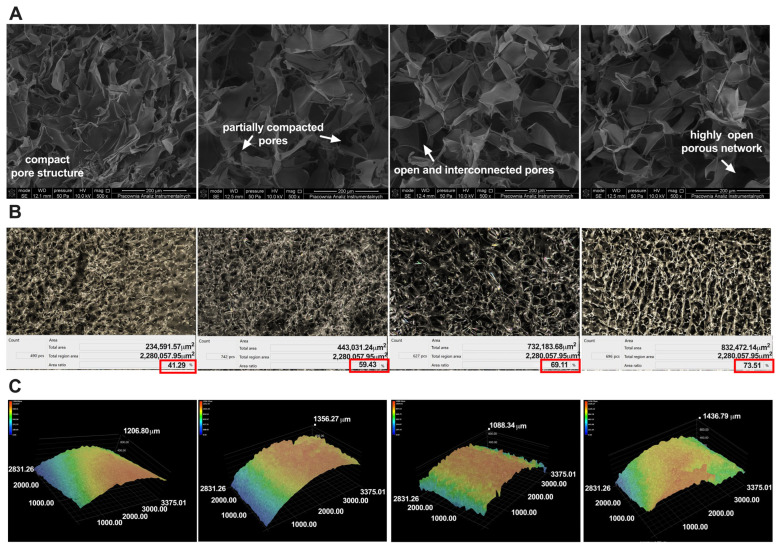
Morphology characterization of the obtained sponges. SEM images (**A**), porosity (**B**), and 3D profile (**C**).

**Figure 3 ijms-26-05314-f003:**
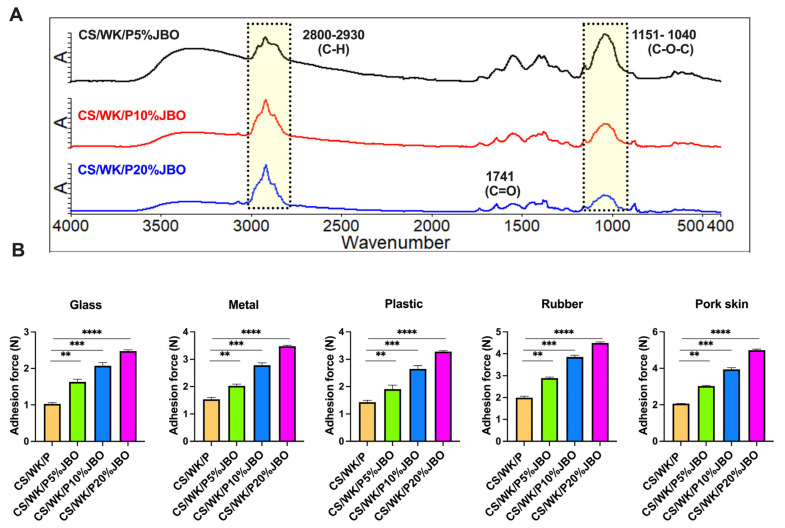
Characterization of the CS/WK/P%JBO sponges. ATR-FTIR spectra (**A**) and adhesion force (**B**). Statistical significance: *p* < 0.001 (**), *p* < 0.0001 (***), *p* < 0.00001 (****).

**Figure 4 ijms-26-05314-f004:**
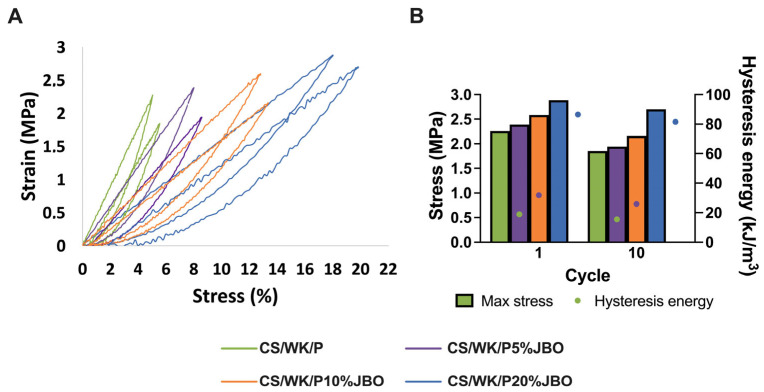
Mechanical cyclic performance of sponges. Strain–stress curves corresponding to the 1st and 10th loading–unloading cycles (**A**), and maximum stress and hysteresis energy recorded for the 1st and 10th cycles (**B**).

**Figure 5 ijms-26-05314-f005:**
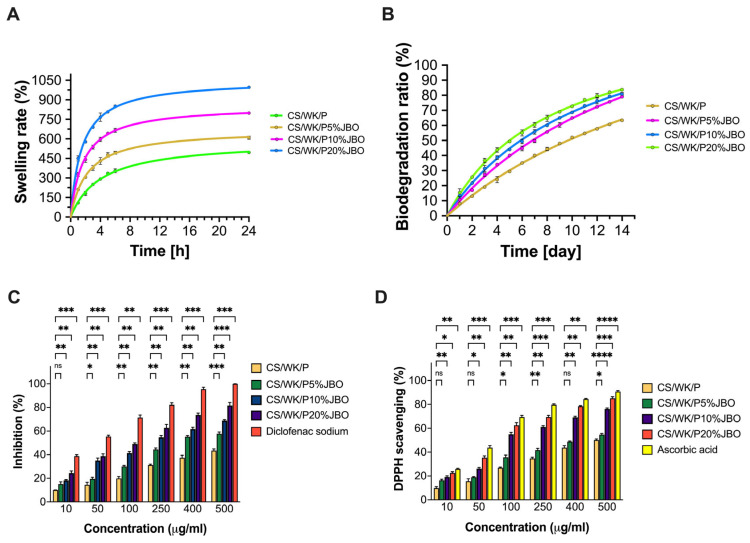
Properties of the sponges. Swelling rate (**A**), biodegradation rate (**B**), anti-inflammatory properties (**C**), and antioxidant properties (**D**). Statistical significance: ns (not significant), *p* < 0.05 (*), *p* < 0.01 (**), *p* < 0.001 (***), *p* < 0.0001 (****).

**Figure 6 ijms-26-05314-f006:**
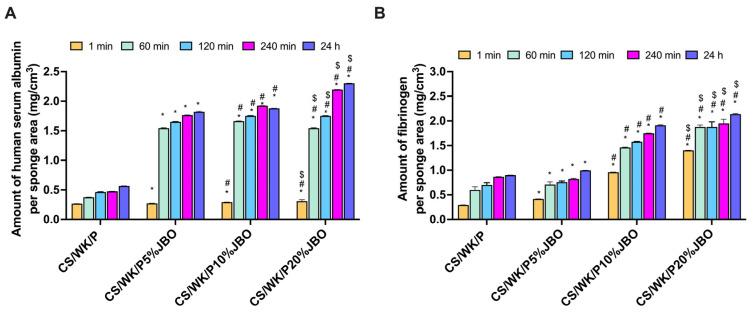
Protein adsorption on the obtained sponges. Amount of the HSA (**A**) and fibrinogen (**B**) bound on the materials. Symbols #, *, and $ indicate *p*  <  0.05 when compared to CS/WK/P, CS/WK/P5%JBO, and CS/WK/P10%JBO, respectively.

**Table 1 ijms-26-05314-t001:** Mechanical properties of the obtained sponges.

Sample	Mechanical Properties		
	Tensile Strength [MPa]	Strain [%]	Young’s Modulus [MPa]
CS/WK/P	2.25 ± 0.05	5.03 ± 0.15	131.2 ± 1.36
CS/WK/P5%JBO	2.39 ± 0.04 ^a^	8.00 ± 0.20 ^a^	120.9 ± 1.37 ^a^
CS/WK/P10%JBO	2.60 ± 0.03 ^a,b^	12.80 ± 0.30 ^a,b^	97.3 ± 0.91 ^a,b^
CS/WK/P20%JBO	2.88 ± 0.05 ^a,b,c^	18.00 ± 0.30 ^a,b^	76.2 ± 1.15 ^a,b,c^

The letters a, b, and c denote statistically significant differences (*p* < 0.05) in comparison to the corresponding CS/WK/P5%, CS/WK/P10%, and CS/WK/P20% sponges.

**Table 2 ijms-26-05314-t002:** Water vapor transmission rate (WVTR) values of the developed sponge compared with representative materials from the literature.

Sample	WVTR (g·m^−2^·day^−1^)	References
CS/WK/P	1950 ± 60	This study
CS/WK/P5%JBO	2020 ± 55 ^a^	This study
CS/WK/P10%JBO	2125 ± 65 ^a,b^	This study
CS/WK/P20%JBO	2250 ± 70 ^a,b,c^	This study
Chitosan, carboxymethyl cellulose, tannic acid	1912.25 ± 13.10	[63]
PVA/PVP, thyme essential oil	606.8 ± 24.2	[64]
PLGA, *Hypericum Perforatum*	1200–3057	[65]
PCL/PVA, *C. majus* extract	2019.82 ± 151.01	[66]
Alginian, chitosan, aloe vera	932.6 ± 9.4065	[67]

The letters a, b, and c denote statistically significant differences (*p* < 0.05) in comparison to the corresponding CS/WK/P5%, CS/WK/P10%, and CS/WK/P20% sponges.

## Data Availability

Data will be made available on request.

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
