# Peer review of "Juniper Berry Oil as a Functional Additive in Chitosan–Water Kefiran–Paramylon Porous Sponges: Structural, Physicochemical, and Protein Interaction Insights"

_ijms, 2025, doi:10.3390/ijms26115314_

Round 1
Reviewer 1 Report
Comments and Suggestions for Authors
- The Introduction section is overly lengthy. Reduce the citation and discussion of previous works, and focus more on highlighting the innovations of this study and the performance advantages of the developed materials (applies to other sections as well).
- In the Results and Discussion section, the claim that "polysaccharide-based materials are non-toxic and biocompatible" lacks theoretical support. Provide references or validate through biocompatibility experiments
- For the properties of juniper berry oil mentioned in Results and Discussion, cite relevant references or conduct experiments to substantiate the claims.
- Redraw Figure 1 to improve visual clarity and explicitly illustrate the preparation steps and mechanisms.
- Include nitrogen adsorption tests to further characterize the pore structure of the material.
- Compare the structure of ginger oil with juniper berry oil to verify whether the increased roughness is due to the presence of the latter.
- In Figure 2 (SEM and porosity), the differences between samples with varying oil content are insufficiently clear. Use arrows or labels to highlight key structural distinctions.
- For the characteristic peaks in Figure 3 (ATR-FTIR, e.g., 2800 cm⁻¹–2930 cm⁻¹), provide literature citations or standard spectral comparisons for peak assignments.
- Elaborate on the material formation mechanism with experimental evidence (e.g., NMR, LC-MS) and structural characterization.
- To assess the material’s potential in biological applications, conduct stress-strain tests to determine maximum load-bearing capacity and cyclic stretching experiments to evaluate fatigue resistance.
- Investigate the impact of increasing JBO content on mechanical properties through comparative experiments.
- Clarify the mechanism by which JBO content affects mechanical properties using experimental evidence rather than speculative descriptions.
- Compare the mechanical enhancement by JBO with similar substances, either through references or additional experiments.
- Expand the adhesion tests to include at least five substrates to demonstrate broader applicability in real-life scenarios.
- Present the material’s WVTRin a table or figure with direct comparisons to same types of materials to highlight its superiority.
- Include direct visual evidence in the anti-inflammatory tests todemonstrate efficacy.
- If feasible, provide experimental validation for the proposed adsorption mechanisms of HSA and Fib.
- Perform statistical analyses (e.g., ANOVA) for Figures 4 and 5 to establish the significance of observed differences.
- The Conclusion mentions biocompatibility, but no cytotoxicity data (e.g., MTT assay) or animal model results are provided. Add in vitro cell experiments.
- Address grammatical inconsistencies (e.g., subject-verb agreement: "The results confirm..." vs. plural subjects) and redundant phrases (e.g., "very unique"). Perform thorough proofreading.
- Standardize unit formats (e.g., g/m²/day" vs. "g·m⁻²·day⁻¹) according to journal guidelines.
Reviewer 2 Report
Comments and Suggestions for Authors
In the article “Juniper berry oil as a functional additive to porous sponges made from chitosan, water, kefiran, and paramylan: structure, physicochemical properties, and interactions with proteins,” the authors evaluated the effect of juniper berry oil on the physicochemical and biological properties of the material obtained.
Point 1
Did the authors replace the lysozyme solution daily during biodegradation?
Point 2
Did the authors specify the maximum amount of juniper berry oil that affects protein adsorption properties?
Author Response
I thank the Editor for considering our manuscript and the Reviewer for reading and reviewing the article. The comments and suggestions we received are valuable and allowed us to improve the quality of the submitted work. Responses to the Reviewer's comments point by point are below. I marked the changes in the text with green color.
- Did the authors replace the lysozyme solution daily during biodegradation?
Thank you for your question. Yes, the lysozyme solution was replaced daily throughout the biodegradation study to ensure consistent enzymatic activity and conditions similar to physiological conditions. This information has been clarified in the “Materials and Methods” section.
P.15. L.519-520: The enzyme-containing PBS solution was replaced daily to maintain enzymatic activity and physiological relevance of the degradation conditions.
- Did the authors specify the maximum amount of juniper berry oil that affects protein adsorption properties?
Thank you for your comment. As part of the research, I also evaluated sponges containing 30% juniper oil, but I did not observe any significant differences in the adsorption levels of HSA and fibrinogen compared to the sample with 20% JBO. Similarly, other physicochemical parameters analyzed (including swelling, degradation, and antioxidant properties) did not show further changes when the oil concentration was increased to 30%. Therefore, for practical and material reasons, I decided not to continue the analysis for higher concentrations and adopted 20% JBO as the optimal concentration for the rest of the study.

Round 2
Reviewer 1 Report
Comments and Suggestions for Authors
no comments